

# Cellular prion protein and γ-synuclein overexpression in LS 174T colorectal cancer cell drives endothelial proliferation-to-differentiation switch

Sing-Hui Ong[1], Kai-Wey Goh[2], Cornelius Kwang-Lee Chieng[1] and Yee-How Say[1]

[1] Department of Biomedical Science, Faculty of Science, Universiti Tunku Abdul Rahman (UTAR) Kampar Campus, Kampar, Perak, Malaysia
[2] Department of Engineering and Science, Centre for Foundation Studies, Universiti Tunku Abdul Rahman (UTAR) Kampar Campus, Kampar, Perak, Malaysia

Corresponding author
Yee-How Say, sayyh@utar.edu.my

## ABSTRACT

**Background:** Tumor-induced angiogenesis is an imperative event in pledging new vasculature for tumor metastasis. Since overexpression of neuronal proteins gamma-synuclein (γ-Syn) and cellular prion protein (PrP$^C$) is always detected in advanced stages of cancer diseases which involve metastasis, this study aimed to investigate whether γ-Syn or PrP$^C$ overexpression in colorectal adenocarcinoma, LS 174T cells affects angiogenesis of endothelial cells, EA.hy 926 (EA).

**Methods:** EA cells were treated with conditioned media (CM) of LS 174T-γ-Syn or LS 174T-PrP, and their proliferation, invasion, migration, adhesion and ability to form angiogenic tubes were assessed using a range of biological assays. To investigate plausible background mechanisms in conferring the properties of EA cells above, nitrite oxide (NO) levels were measured and the expression of angiogenesis-related factors was assessed using a human angiogenesis antibody array.

**Results:** EA proliferation was significantly inhibited by LS 174T-PrP CM whereas its telomerase activity was reduced by CM of LS 174T-γ-Syn or LS 174T-PrP, as compared to EA incubated with LS 174T CM. Besides, LS 174T-γ-Syn CM or LS 174T-PrP CM inhibited EA invasion and migration in Boyden chamber assay. Furthermore, LS 174T-γ-Syn CM significantly inhibited EA migration in scratch wound assay. Gelatin zymography revealed reduced secretion of MMP-2 and MMP-9 by EA treated with LS 174T-γ-Syn CM or LS 174T-PrP CM. In addition, cell adhesion assay showed lesser LS 174T-γ-Syn or LS 174T-PrP cells adhered onto EA, as compared to LS 174T. In tube formation assay, LS 174T-γ-Syn CM or LS 174T-PrP CM induced EA tube formation. Increased NO secretion by EA treated with LS 174T-γ-Syn CM or LS 174T-PrP CM was also detected. Lastly, decreased expression of pro-angiogenic factors like CXCL16, IGFBP-2 and amphiregulin in LS 174T-γ-Syn CM or LS 174T-PrP CM was detected using the angiogenesis antibody array.

**Discussion:** These results suggest that overexpression of γ-Syn or PrP$^C$ could possibly be involved in colorectal cancer-induced angiogenesis by inducing an endothelial proliferation–differentiation switch. NO could be the main factor

in governing this switch, and modulation on the secretion patterns of angiogenesis-related proteins could be the strategy of colorectal cancer cells overexpressing γ-Syn or PrP$^C$ in ensuring this transition.

# INTRODUCTION

Angiogenesis, the formation of new capillaries from pre-existing blood vessels, plays a critical role in a wide range of physiological and pathological events like embryonic development, wound healing, cancer, rheumatoid arthritis, ischemic heart disease and atherosclerosis (*Felmeden, Blann & Lip, 2003*). Angiogenesis mechanisms involve complex and diverse cellular actions, such as proliferation, migration, and morphological differentiation of endothelial cells (*Bussolino, Mantovani & Persico, 1997*). All of these events are strictly controlled by many naturally occurring pro- and anti-angiogenic molecules (*Ye, 2016*).

γ-Synuclein (γ-Syn) and cellular prion protein (PrP$^C$), predominantly expressed in neurons of the brain and implicated in neurodegenerative diseases (*Lavedan et al., 1998*; *Prusiner, 1998*), are found to be aberrantly overexpressed in colorectal cancer (*Liu et al., 2005*; *Antonacopoulou et al., 2008*). Substantial studies illustrate the role of these proteins in fostering tumorigenesis, including promoting cancer cell migration, adhesion, invasion, and survival (*Liu et al., 2005*; *Ye et al., 2009*; *Yap & Say, 2011*, *2012*). Previously, we showed that the overexpression of γ-Syn in LS 174T colorectal adenocarcinoma cell line could confer both pro-invasive and doxorubicin-mediated pro-apoptotic properties to the cells (*Goh & Say, 2015*). In addition, we also showed that PrP$^C$ could promote LS 174T cells carcinogenesis by increasing invasiveness and resistance against doxorubicin-induced apoptosis (*Chieng & Say, 2015*).

While there is growing literature on the role of γ-Syn and PrP$^C$ in conferring tumor aggressiveness, the role of these proteins in angiogenesis is still elusive. Hence, in this study, we aimed to investigate the capability of LS 174T cells overexpressing γ-Syn or PrP$^C$ in modulating human immortalized endothelial cells, EA.hy 926 angiogenic response, by incubating EA.hy 926 with the conditioned media of LS 174T cells overexpressing γ-Syn or PrP$^C$. In this study, we demonstrated that overexpression of PrP$^C$ and γ-Syn in LS 174T cells drives EA.hy 926 into a proliferation-to-differentiation switch, as evidenced by reduced proliferation, invasion, migration and heteroadhesion, but increased tube formation. Increased secretion of nitric oxide (NO) by EA.hy 926 and decreased extracellular secretion of pro-angiogenic factors like CXCL16, IGFBP-2 and amphiregulin by LS 174T overexpressing PrP$^C$ or γ-Syn, are thought to play a role in this switch. This study suggests that neuronal proteins PrP$^C$ and γ-Syn are not only involved in the cancer biology of colorectal cancer cells, but are also involved in the tumor microenvironment by modulating tube formation of endothelial cells nearby.

## MATERIALS AND METHODS

### Cell culture and treatments

LS 174T (ATCC® CL-188™) and EA.hy 926 (ATCC® CRL-2922™), obtained from the American Type Culture Collection (ATCC), were maintained in Eagle's Minimum Essential Medium (EMEM) (Thermo Fisher Scientific, Waltham, MA, USA) and Dulbecco's Modified Eagle's Medium (DMEM) (Thermo Fisher Scientific, Waltham, MA, USA), respectively, supplemented with 10% (v/v) fetal bovine serum (Sigma-Aldrich, St. Louis, MO, USA) and 1% (v/v) penicillin–streptomycin (Nacalai Tesque, Kyoto, Japan) at 37 °C and 5% $CO_2$ in air. All cell lines have been checked to ensure they are free of contamination and have been used from young stock (less than seven passages). LS 174T overexpressing γ-Syn (LS 174T-γ-Syn) and PrP (LS 174T-PrP) were previously established in our laboratory (*Chieng & Say, 2015*; *Goh & Say, 2015*), and overexpression of γ-Syn and PrP$^C$ in LS 174T cells was confirmed once again in this study by Western blotting, as detailed previously (*Chieng & Say, 2015*; *Goh & Say, 2015*). In this study, untransfected LS 174T, LS 174T-γ-Syn and LS 174T-PrP cells are collectively termed as LS 174T cell lines. The angiogenic effects of LS 174T cell lines conditioned media (CM) were evaluated by incubating EA.hy 926 (EA) with the CM. Treatments of EA with LS 174T cell lines CM are termed as EA-LS CM Tx, EA-LS-γ-Syn Tx and EA-LS-PrP Tx. EA treated with serum-free DMEM (empty medium), EA EM Tx, served as the negative control. Generally, LS 174T cell lines cells were seeded in appropriate culture vessels. The culture media were replaced with serum-free DMEM and the cells were incubated for 24 h (*He et al., 2015*). The CM were then collected and used to treat EA.

### In vitro angiogenesis assays

#### MTT cell proliferation assay

LS 174T cell lines and EA were seeded in 96-well tissue culture plates ($2 \times 10^4$ cells/well). CM of LS 174T cell lines were collected accordingly and transferred to EA. Following CM treatment, EA cell viability was determined at every 24 h for 72 h using 3-(4-5-dimethylthiazol-2-yl)-2,5-diphenyltetrazolium bromide (MTT) (Bio Basic Inc., Markham, Canada) assay, as described previously (*Chieng & Say, 2015*).

#### Endothelial telomerase activity analysis

LS 174T cell lines and EA were seeded in $T_{25}$ tissue culture flasks ($1 \times 10^6$ cells/flask). CM of LS 174T cell lines were collected accordingly and transferred to EA. After 24 h incubation, telomerase activity in treated EA cells was analyzed using TRAPeze® Telomerase Detection Kit (Merck Millipore, Billerica, MA, USA), according to the manufacturer's protocol.

#### Boyden chamber assay

Boyden chamber assay was performed using the Cultrex® In Vitro Angiogenesis Assay Endothelial Cell Invasion Kit (Trevigen, Gaithersburg, MD, USA). Briefly, EA ($2 \times 10^4$ cells) were loaded onto the top chamber wells of Boyden chamber, with (invasion) or without (migration) prior coating with basement extracellular membrane (BME)

on the 8 $\mu$m polyethylene terephthalate (PET) membrane in the well. CM from LS 174T cell lines cultured in 96-well tissue culture plate ($2 \times 10^4$ cells/well) were loaded into the lower chamber wells of Boyden chamber, and the plate was incubated in cell culture incubator for 24 h. Successfully invaded and migrated cells were stained with Calcein AM and fluorescence intensity was quantified with Infinite 200 PRO® microplate reader (Tecan, Männedorf, Switzerland) using 485/530 nm excitation/emission wavelengths.

### Gelatin zymography

LS 174T cell lines and EA were seeded in $T_{25}$ tissue culture flasks ($1 \times 10^6$ cells/flask). CM of LS 174T cell lines were collected accordingly and transferred to EA. Cell lysates and CM of treated EA were collected after 24 h incubation. CM of LS 174T cell lines were also analyzed. Samples were subjected to protein quantification and gelatin zymography was performed as described previously (*Gao et al., 2011*).

### Scratch wound assay

EA cells were seeded in 24-well tissue culture plates ($1 \times 10^5$ cells/well). A yellow pipette tip (P-100) was used to introduce a vertical "wound" by clearing an area of the confluent EA monolayer. Prior to this, CM of LS 174T cell lines were collected from cells cultured in 24-well tissue culture plates ($1 \times 10^5$ cells/well) and transferred to wounded EA monolayer. Images were taken every 3 h for 12 h with Eclipse TS100 inverted microscope (Nikon, Melville, NY, USA) at $100\times$ magnification and the percentage of remaining cleared or unfilled area was evaluated with TScratch™ software Version 1.0 by CSE lab (http://cse-lab.ethz.ch/software/).

### Cell adhesion assay

Cell adhesion assay was carried out by referring to the protocol of the Chemicon® Endothelial Cell Adhesion Assay Kit (Merck Millipore, Billerica, MA, USA). Briefly, LS 174T cell lines ($1 \times 10^6$ cells/$T_{25}$ flask) unstimulated or stimulated with vascular endothelial growth factor (VEGF) (Sigma-Aldrich, St. Louis, MO, USA) at 20 $\mu$g/mL for 4 h were trypsinized, centrifuged, and cell pellets were mixed with Calcein AM (Trevigen, Gaithersburg, MD, USA). The mixture was incubated in cell culture incubator for 30 min. Next, cells were pelleted down at 400$g$ for 5 min and supernatant was discarded. Cells were then washed with PBS by gentle suspension and subjected to centrifugation again. This washing step was repeated for three times to remove excess Calcein AM. After the last wash, cells were resuspended with DMEM and $2 \times 10^4$ cells/well were loaded on top of confluent monolayer EA cells ($2 \times 10^4$ cells/well) in a 96-well plate. After 1 h incubation, media from wells were gently aspirated and discarded. Cells were washed once with PBS to remove unbound LS 174T cells. Fluorescent-labeled cells that adhered on EA were quantified with Infinite 200 PRO® microplate reader (Tecan, Männedorf, Switzerland) using 485/530 nm excitation/emission wavelengths.

### Endothelial tube formation assay

Endothelial tube formation assay was carried out using Cultrex® In Vitro Angiogenesis Assay Tube Formation Kit (Trevigen, Gaithersburg, MD, USA) according to the

manufacturer's protocol. Briefly, CM of LS 174T cell lines were collected accordingly, mixed with EA cell suspension, and $2 \times 10^4$ cells/well were plated on 96-well tissue culture plate coated with Matrigel. After 8 h of incubation, images were acquired using Eclipse TS100 inverted microscope (Nikon, Melville, NY, USA) at 100× magnification. Extent of the tube formation—cell covered area, total loops, total branching points, total tube length and total tubes—was quantified by analyzing photographed images with WimTube™ software by Wimasis Image Analysis Platform (https://www.wimasis.com/en/products/13/WimTube).

### Nitrite oxide quantification

LS 174T cell lines and EA were seeded in 96-well tissue culture plates ($2 \times 10^4$ cell/well). CM of LS 174T cell lines were collected accordingly and transferred to EA. Following incubation, NO level in media was quantified using Griess reagent. Equal volume of Griess reagent (A: 0.1% $N$-(1-naphthyl)ethylenediamine dihydrochloride; B: 1% sulfanilamide in 5% phosphoric acid; A:B=1.1) and media were mixed together and incubated at room temperature for 15 min. The absorbance value was taken at 540 nm using Infinite 200 PRO® microplate reader (Tecan, Männedorf, Switzerland).

### Partial secretome analysis of LS 174T CM

LS 174T cells were seeded in six-well tissue culture plate ($7.5 \times 10^5$ cells/well). Media were replaced with 1.5 mL of serum-free DMEM and incubated for 24 h. CM were processed accordingly and concentrated with vacuum concentrator (Scanvac, Allerød, Denmark). CM were analyzed with Proteome Profiler™ Human Angiogenesis Array Kit (ARY007; R&D Systems, Minneapolis, MN, USA) according to the manufacturer's protocol. This kit enables the simultaneously detection of the relative levels of 55 angiogenesis-related proteins in a single sample. Results were captured with ChemiDoc™ XRS System (Biorad, Hercules, CA, USA) and analyzed with NIH ImageJ image processing and analysis software.

### Statistical analysis

Results were presented as mean ± standard error of the mean (SEM) of at least two independent experiments, performed in at least triplicates, unless otherwise stated. Statistical analysis was performed using one-way ANOVA followed by LSD's post hoc test for multiple comparisons using IBM SPSS Statistics software version 16.0 (IBM, New York, NY, USA). A $p$-value of less than 0.05 was considered as statistically significant.

## RESULTS

### LS 174T-PrP CM reduces EA proliferation whereas both LS 174T-γ-Syn CM and LS 174T-PrP CM reduce EA telomerase activity

Western blot confirmed the overexpression of γ-Syn and PrP$^C$ in LS 174T-γ-Syn and LS 174T-PrP cells, respectively, as established previously (*Chieng & Say, 2015*; *Goh & Say, 2015*) (Figs. 1A and 1B, respectively). MTT proliferation assay was then carried out to evaluate the effects of LS 174T CM on EA proliferation rate and to determine the optimal

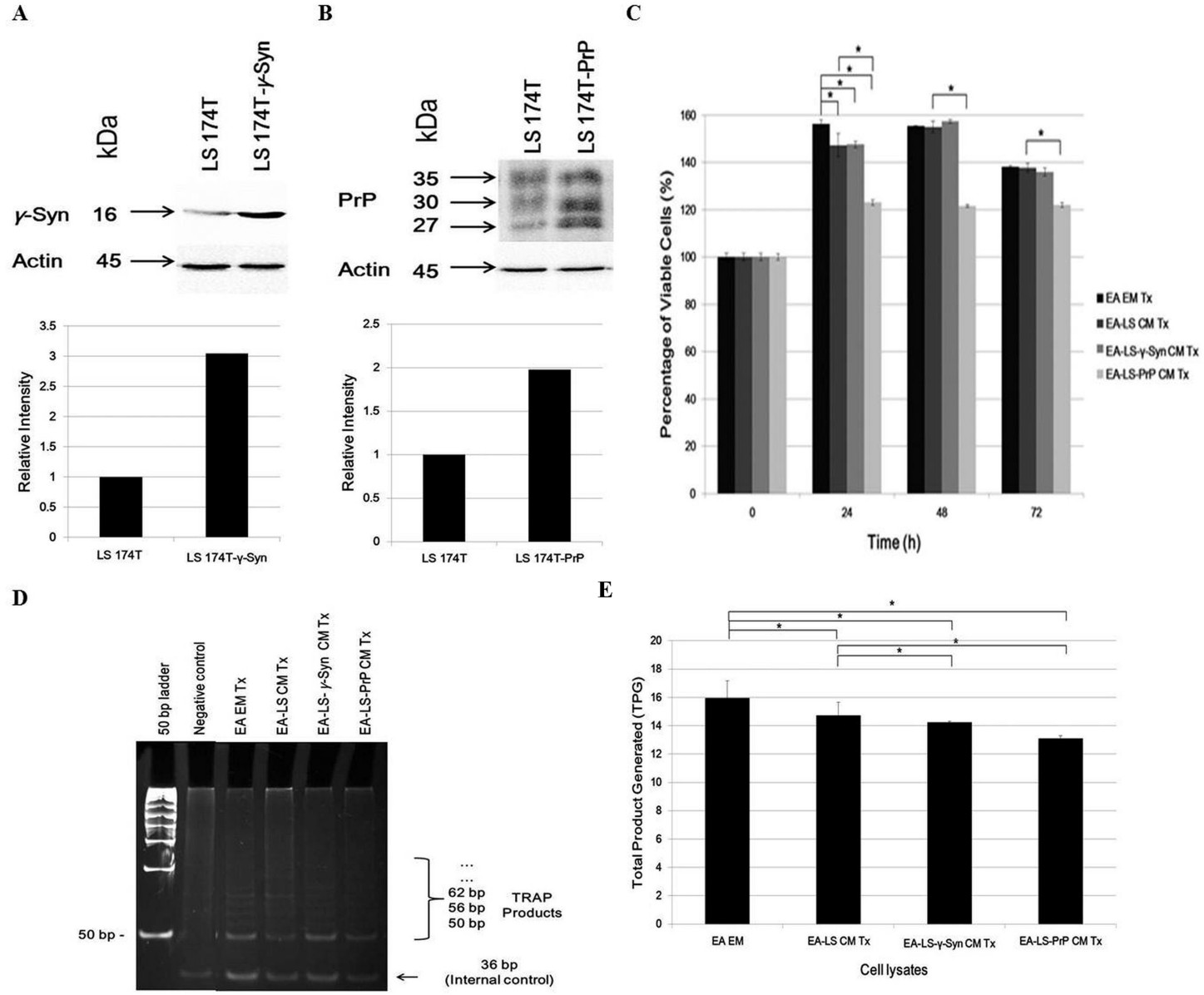

**Figure 1  LS 174T-PrP CM reduces EA proliferation whereas LS 174T-γ-Syn CM or LS 174T-PrP CM reduces EA telomerase activity.** Western blot and densitometry analyses of (A) γ-Syn and (B) PrP overexpression in LS 174T cells transfected with SNCG (LS 174T-γ-Syn) and PRNP (LS 174T-PrP) genes. Band densitometry was quantified using ImageJ and the densitometry values of γ-Syn or PrP were divided by those of actin; LS 174T was set at 1 and the relative intensities of LS 174T-γ-Syn or LS 174T-PrP were calculated and plotted. (C) Effects of LS 174T cell lines (untransfected LS 174T, LS 174T-γ-Syn and LS 174T-PrP) CM on EA proliferation rate as determined by MTT proliferation assay. (D) PCR analysis of telomerase activity in cell lysates of EA treated with LS 174T cell lines CM (EA-LS CM Tx, EA LS-γ-Syn CM Tx and EA-LS-PrP CM Tx). EA treated with empty medium (EA EM Tx) served as a negative control. A 36 bp band (S-IC) internal control is produced from oligonucleotides K1 and TSK1 (provided in kit) together with substrate oligonucleotide. It serves to monitor PCR inhibition, a control for amplification efficiency in each reaction and is used for quantitative analysis of the reaction products. (E) Total product generated (TPG) was determined by analyzing the density of TRAP products generated from each sample using ImageJ software, and calculated according to the manufacturer's protocol. Data represent the mean ± SEM (error bars) of three independent experiments. Mean values were compared using one-way ANOVA followed by LSD's post hoc test. Asterisk indicates $p < 0.05$.

treatment duration for the rest of the study (Fig. 1C). At 24 h incubation, LS 174T cell lines CM suppressed EA proliferation rate as compared to EA EM Tx. In addition, LS 174T-PrP CM exerted strong suppression on EA proliferation rate whereas LS 174T-γ-Syn had no effect, as compared to EA-LS CM Tx. The effects of LS 174T cell lines CM on EA proliferation rate at 48 h were comparable to that of 24 h, and further incubation till 72 h showed an overall reduction in EA viability. Thus, 24 h incubation was chosen to be the standard treatment duration.

Reactivation of telomerase expression is implicated in cancer cell transformation, which supports the uncontrolled replication of cancer cells (*Shay & Wright, 2011*). Therefore, we investigated if the overexpression of γ-Syn or PrP$^C$ in colorectal cancer cells would also affect the expression or activity of telomerase in endothelial cells. We hypothesized that telomerase activity in EA treated with LS 174T CM would be reduced since the CM did not support EA proliferation. Indeed, telomerase activity in EA treated with LS 174T cell lines was reduced, as compared to EA EM Tx, with EA-LS-PrP CM Tx yielding the lowest total product generated (TPG), reflecting telomerase activity (Figs. 1D and 1E). Taken together, overexpression of γ-Syn or PrP$^C$ in LS 174T cells do not support EA proliferation.

## LS 174T-γ-Syn CM and LS 174T-PrP CM do not promote EA invasiveness and migration

The angiogenic effects of LS 174T cell lines CM were also being evaluated for its capability in modulating EA invasiveness and migration. Boyden chamber assay revealed that CM of LS 174T-γ-Syn and LS174T-PrP reduced the invasiveness and migration of EA, as compared to EA-LS CM Tx (Fig. 2A).

To confirm this phenomenon, we evaluated the expression of matrix metalloprotease-2 and -9 (MMP-2 and MMP-9), two gelatinases which facilitate cell invasion by breaking down the extracellular matrix (*Said, Raufman & Xie, 2014*). Gelatin zymography revealed only MMP-2 was expressed in EA, similar with a previous finding (*Wu, Lin & Chen, 2005*). Although MMP-2 expression was upregulated in EA treated with LS 174T cell lines CM as compared to EA EM Tx, its expression was downregulated in EA-LS-PrP CM Tx (Fig. 2B). Besides, LS 174T cell lines secreted low levels of MMP-2 and the secretion of MMP-9 from LS 174T-PrP cells was the lowest (Fig. 2C). When EA were treated with LS 174T cell lines CM, reduced MMP-2 and MMP-9 secretion by EA cells treated with LS 174T-γ-Syn CM or LS 174T-PrP CM was observed, as compared to treatment with LS 174T CM (Fig. 2C). Although the modulation was not significant, the accordance with Boyden chamber assay results suggests the possible role of γ-Syn and PrP$^C$ in LS 174T cells in inhibiting EA invasion.

Furthermore, scratch wound assay was performed to further confirm the role of overexpressed γ-Syn or PrP in LS 174T cells in affecting EA migration (Fig. 2D). At 12 h incubation, LS 174T-γ-Syn CM strongly inhibited EA migration, conforming the Boyden chamber assay results.

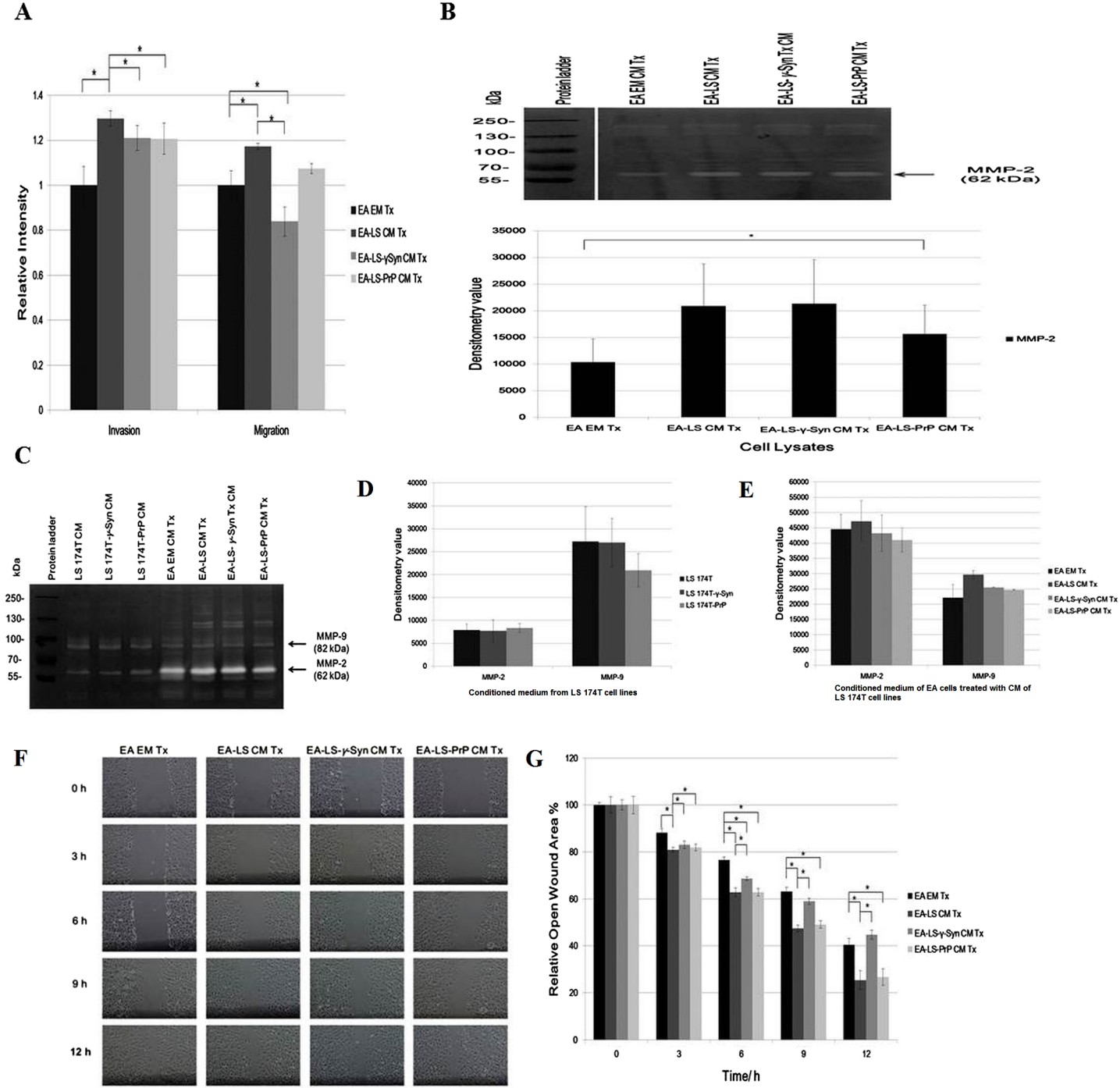

**Figure 2 LS 174T-γ-Syn CM and LS 174T-PrP CM do not promote EA invasiveness and migration.** (A) Effects of LS 174T cell lines CM on EA invasiveness and migration were evaluated with Boyden chamber assay. Successfully invaded and migrated cells were stained with Calcein AM and fluorescence intensity was quantified at 485/530 nm excitation/emission wavelengths. (B) Gelatin zymography and densitometry analyses (by ImageJ) of MMP-2 and MMP-9 expression in EA cell lysates treated with LS 174T cell lines CM. (C) Gelatin zymography and densitometry analyses (by ImageJ) of MMP-2 and MMP-9 secretion by LS 174T cell lines (D) and EA treated with CM of LS 174T cell lines (E). (F) Scratch wound assay revealed the effects of LS 174T cell lines CM on EA motility. Images were taken at 100× magnification using the Eclipse TS100 inverted microscope (Nikon, New York, NY, USA). (G) The remaining open wound area of each sample at different time points was analyzed using TScratch™ software, and presented in the bar chart of percentage relative open wound vs. time (h). Data represent the mean ± SEM (error bars) of three independent experiments. Mean values were compared using one-way ANOVA followed by LSD's post hoc test. Asterisk indicates $p < 0.05$.

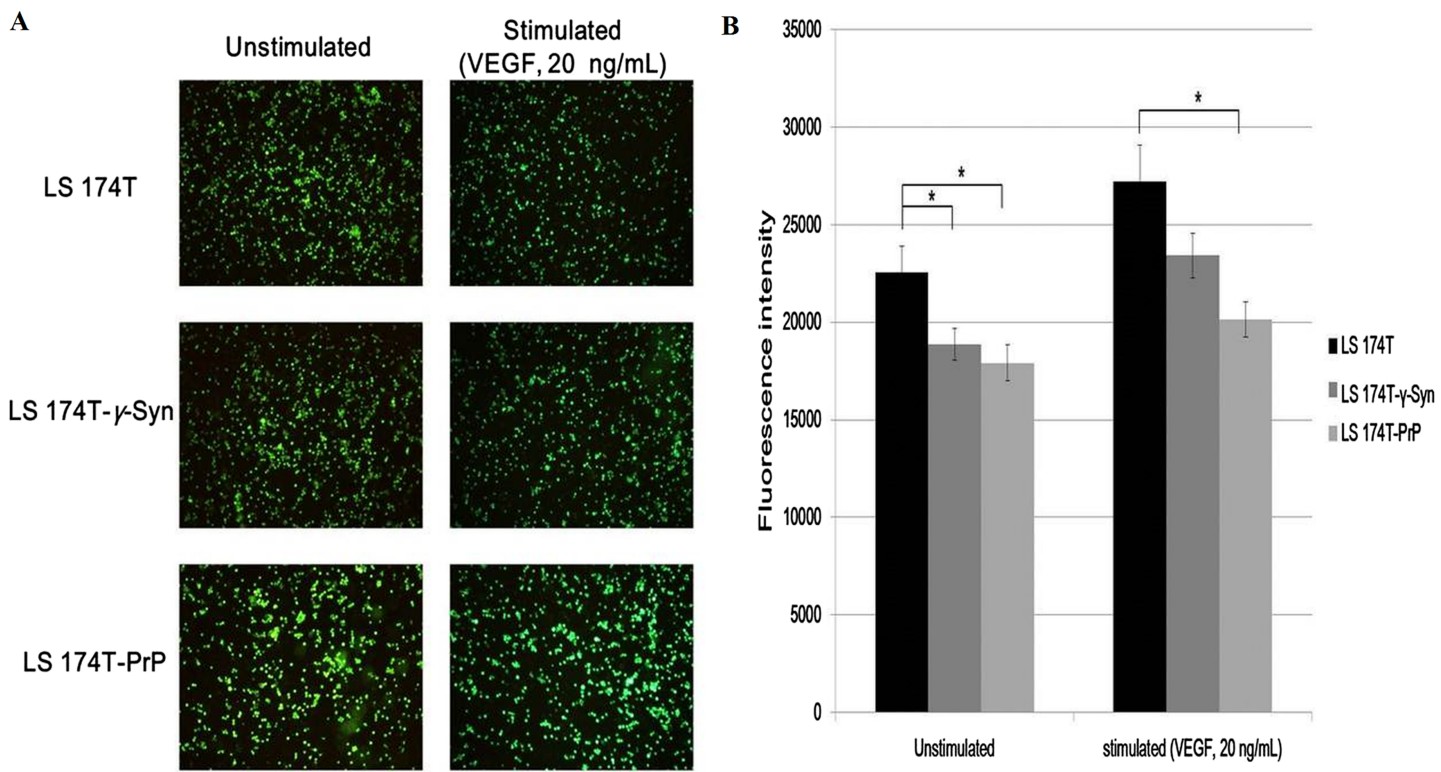

**Figure 3 Overexpression of γ-Syn and PrPC inhibits LS 174T cells from adhering onto EA.** (A) Cell attachment analysis of the adhesiveness of unstimulated and stimulated (VEGF, 20 ng/mL) LS 174T cell lines on EA. Images were taken at $100\times$ magnification using the Eclipse TS100 inverted microscope (Nikon, New York, NY, USA). (B) Fluorescence intensity was quantified. Data were expressed as fluorescence intensity and represent the mean ± SEM (error bars) of three independent experiments. Mean values were compared using one-way ANOVA followed by LSD's post hoc test. Asterisk indicates $p < 0.05$, as compared to LS 174T.

## Overexpression of γ-Syn or PrP$^C$ inhibits LS 174T cells from adhering onto EA

Adhesion of cancer cells on endothelial cells is a crucial event in facilitating cancer metastasis (*Golias et al., 2005*). We investigated if the overexpression of γ-Syn and PrP$^C$ in LS 174T cells would aid the adhesion of cancer cells onto EA. Cell adhesion assay showed that γ-Syn or PrP overexpression did not promote the adhesion of LS 174T cells onto EA, as compared to untransfected LS 174T cells (Fig. 3). Furthermore, although prior stimulation of LS 174T cell lines cells with VEGF—a well-documented pro-adhesion angiogenic factor (*Bendas & Borsig, 2012*)—increased the bound LS 174T cell lines cells on EA, this factor did not successfully counteract the inhibitory effects of γ-Syn or PrP$^C$ overexpression (Fig. 3).

## LS 174T-γ-Syn CM and LS 174T-PrP CM induce EA tube formation, possibly by inducing NO secretion from EA

Formation of new vasculature is considered as complete when the proliferated and migrated endothelial cells successfully differentiate and form blood vessels (*Bussolino, Mantovani & Persico, 1997*). As EA is an immortalized cell line, it has better stability
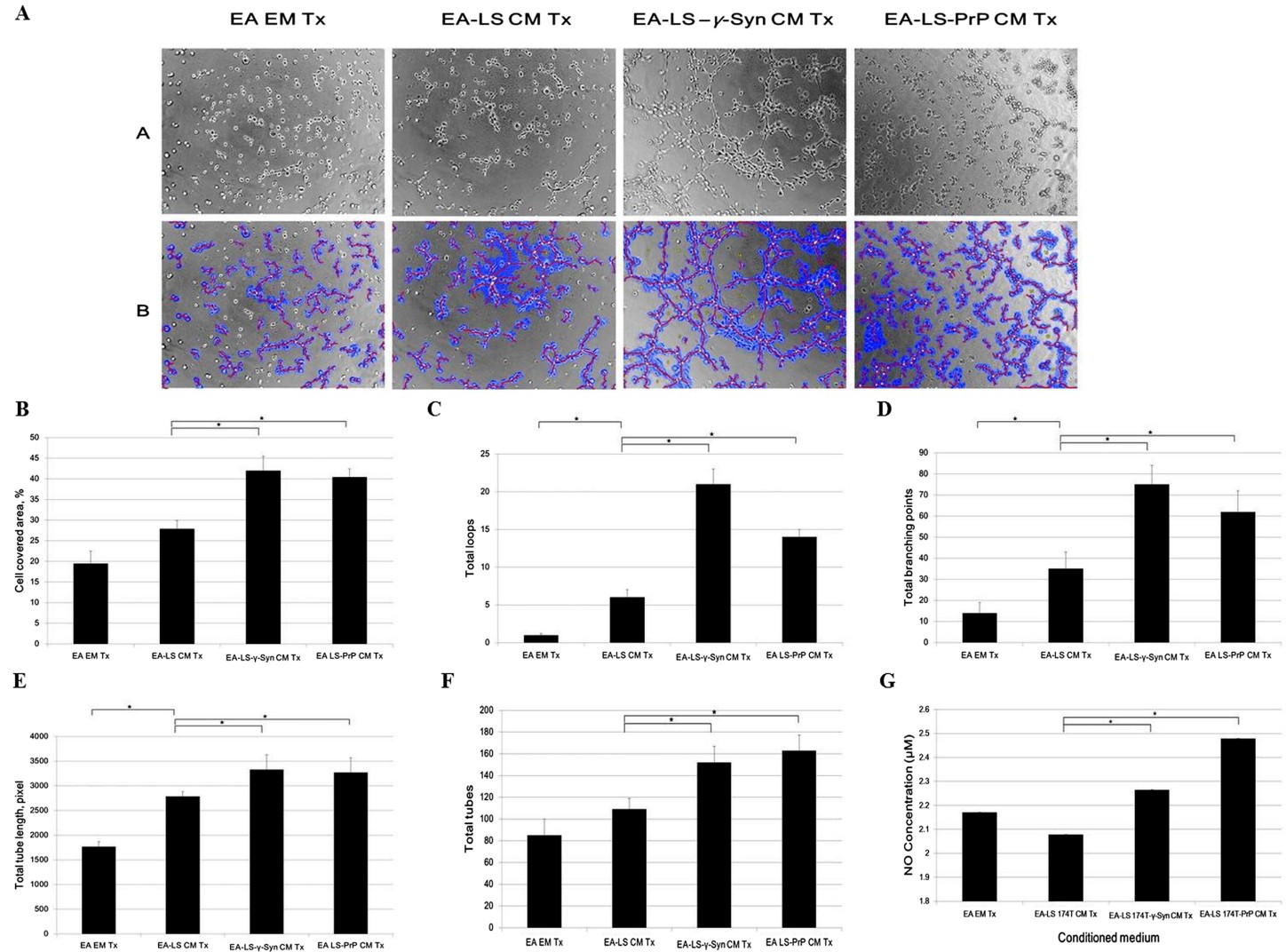

**Figure 4 LS 174T-γ-Syn CM or LS 174T-PrP CM induces EA tube formation, possibly by inducing NO secretion from EA.** (A) Panel A shows the effects of LS 174T cell lines CM on the differentiation and tube formation of EA.hy 926. Images were taken at 100× magnification using the Eclipse TS100 inverted microscope (Nikon, New York, NY, USA); Panel B shows the analyzed images with WimTube[TM] software by Wimasis Image Analysis Platform. (B–F) Extent of EA tube formation upon treatment with LS 174T cell lines CM was evaluated through cell covered area, total loops, total branching points total tube length and total tubes. (G) Griess assay was performed to determine NO secretion level of EA treated with CM of LS 174T cell lines. Data represent the mean ± SEM (error bars) of three independent experiments. Mean values were compared using one-way ANOVA followed by LSD's post hoc test. Asterisk indicates $p < 0.05$.

through passage number and better reproducibility of results while still retaining the characteristics of human vascular endothelium (*Edgell, McDonald & Graham, 1983*). It has been used as a model for in vitro tube formation angiogenesis assay (*Aranda & Owen, 2009*). Therefore, we evaluated the ability of LS 174T cell lines CM in inducing EA tube formation. We showed that EA treated with LS 174T-γ-Syn or LS 174T PrP CM successfully formed tube-like structures (Fig. 4A). WimTube[TM] software analyses revealed increase in cell covered area, total loops formed, total branching points, total tube length

and total tubes formed (Figs. 4B–4F), which suggest the pro-tube formation effect of LS 174T-γ-Syn and LS 174T PrP CM.

Up to this point, we observed a switch from proliferation to differentiation in EA cells. We further elucidated this phenomenon by evaluating the secretion of a possible factor—NO—which has been shown to terminate the proliferative actions of angiogenic growth factors and promoted EC differentiation into vascular tubes in the angiogenic response to basic fibroblast growth factor (bFGF) (Babaei et al., 1998). Indeed, both LS 174T-γ-Syn CM and LS 174T-PrP CM increased the secretion of NO from EA (Fig. 4G).

## Partial secretome profiling of CM from LS 174T cell lines reveals differential regulation of angiogenic factors

Investigation of the partial secretome of CM from LS 174T cell lines for the expression of angiogenic factors was achieved by performing an antibody array assay with 55 angiogenesis-related proteins (Fig. 5A). Twenty one targets showed detectable signals in the array blot (Fig. 5B) and out of these, ten angiogenic factors which gave strong differential expression were relatively quantified compared to LS 174T CM (Fig. 5C). Chemokine (C–X–C motif) ligand 16 (CXCL16), dipeptidyl peptidase-4 (DPPIV), insulin growth factor binding protein-2 (IGFBP-2) and amphiregulin secretions were downregulated in LS 174T-γ-Syn and LS 174T-PrP CM, compared to LS 174T CM, with CXCL16, DPPIV, and amphiregulin showing significance in LS 174T-PrP CM. The secretion of interleukin-8 (IL-8) from LS 174T-γ-Syn was exceptionally heightened, whereas moderate secretions of *urokinase*-type plasminogen activator (uPA) and angiogenin were found in LS 174T-PrP.

## DISCUSSION

Continuing our research in elucidating the roles of neuronal proteins γ-Syn and PrP$^C$ in colorectal cancer cell biology (Yap & Say, 2011, 2012; Chieng & Say, 2015; Goh & Say, 2015), we investigated the angiogenic effects of these two proteins in LS 174T cells by performing in vitro angiogenesis assay. Angiogenesis is a complex process which involves the proliferation, invasion, migration, and tube formation of ECs (Bussolino, Mantovani & Persico, 1997; Falchetti et al., 2008). In our study, LS 174T-PrP CM reduced EA proliferation whereas both LS 174T-γ-Syn CM and LS 174T-PrP CM reduced EA telomerase activity. This result is supported by the work of Falchetti et al. (2008), where they showed inhibition of telomerase disrupted the proliferation of ECs.

Before ECs could gain access to a tumor site to form new vasculature, it must first invade the basement membrane, a specialized structure or barrier which separates parenchymal cells from stromal tissues (Mylonas & Lazaris, 2014). Boyden chamber assay showed that overexpression of γ-Syn or PrP$^C$ did not promote EA invasion and migration. Up to date, no studies depict the role of these two neuronal proteins on ECs invasiveness and migration ability. Therefore, we confirmed our observation by performing gelatin zymography and scratch wound assay. The accordance of these results

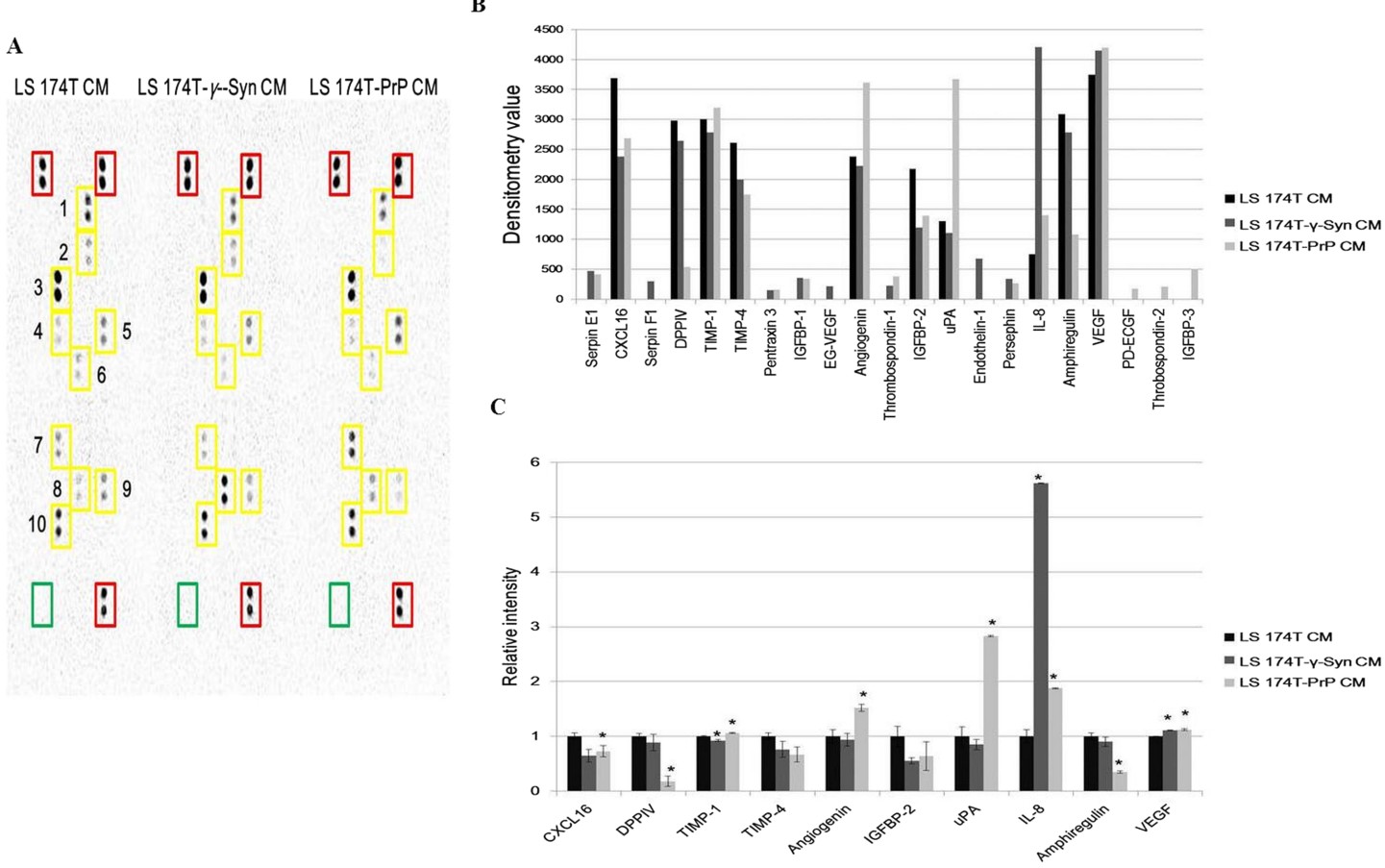

**Figure 5** **Partial secretome analysis of conditioned media from LS 174T cell lines.** (A) Angiogenesis antibody array proteome profiling of CM of LS 174T cell lines. Red boxes indicate reference spots whereas green boxes indicate negative control. Numbered yellow boxes indicate the ten angiogenic factors in the CM that yielded distinct signals, which are CXCL16 (1), DPPIV (2), TIMP-1 (3), TIMP-4 (4), angiogenin (5), IGFBP-2 (6), uPA (7), IL-8 (8), amphiregulin (9) and VEGF (10). (B) Secretion levels of different angiogenesis-related proteins in LS 174T CM, LS 174T-γ-Syn CM and LS 174T-PrP CM were analyzed with ImageJ software and presented as densitometry values. (C) The ten angiogenic factors that yielded distinct signals were relatively quantified. Data of densitometry value were expressed as relative intensity compared to LS 174T CM (set as 1) and represent the mean ± SEM (error bars) of two independent experiments. Mean values were compared using one-way ANOVA followed by LSD's post hoc test. Asterisk indicates $p < 0.05$ as compared to LS 174T CM.

to the Boyden chamber assay results suggests that γ-Syn and PrP[C] do not support EA invasion and migration, possibly by downregulating the expression of MMP-2 and MMP-9.

The adhesive contact between tumor cells and ECs (cell–cell) is particularly important in the extravasation of tumor cells from the circulation to a distant tissue, and eventually proliferation at the secondary site (*Bendas & Borsig, 2012*). Our results showed that overexpression of γ-Syn or PrP[C] in LS 174T cells did not promote the adhesion of the cells on EA. Stimulation of LS 174T cell lines with VEGF enhanced the adhesion of LS 174T cell lines on EA but did not rescue the hampering effects of overexpressed γ-Syn or PrP[C].

The last step in the establishment of new vasculature is the ability of ECs to undergo morphogenesis, a state where ECs differentiate to form vascular tubes. In in vitro assay,
this happens when EC rearranges, aligns and elongates to form capillary-like structures that subsequently interact with each other to form extensive interconnecting networks (*Davis & Camarillo, 1996*). In our study, overexpression of γ-Syn or PrP$^C$ promoted the differentiation and morphogenesis of ECs. γ-Syn has been documented earlier to play a role in the neovascularization and subsequent growth of human endometriotic lesions (*Edwards et al., 2014*). Although the role of PrP$^C$ in angiogenesis is unknown, its paralog—doppel, has been shown to promote tumor angiogenesis via VEGF signaling (*Al-hilal et al., 2016*). As a matter of fact, PrP$^C$ has been recognized as an important factor in the differentiation of human embryonic stem cells into neuron-, oligodendrocyte-, and astrocyte-committed lineages (*Lee & Baskakov, 2014*). These suggest that γ-Syn and PrP$^C$ are involved in endothelial differentiation and morphogenesis and might be involved in tumor angiogenic tube formation, particularly in colorectal cancer.

Nitric oxide displays a biphasic role in carcinogenesis and tumor progression, depending on its concentration; low NO concentration (<100 nM) has pro-tumor effect whereas at high concentration (>500 nM), it could induce cytostasis (*Napoli et al., 2013*). NO has also been shown to terminate the proliferative actions of angiogenic growth factors and promoted EC differentiation into vascular tubes in response to exogenous bFGF addition (*Babaei et al., 1998*). In our study, we observed reduced EA proliferation rate, invasion and migration, and reduced EA heteroadhesion to colon cancer cells, but successful EA tube formation, upon treatment with LS 174T-γ-Syn CM or LS 174T-PrP CM. Therefore, we speculate that the micromolar secretion of NO by EA cells upon treatment with LS 174T overexpressing γ-Syn or PrP$^C$, is partially responsible for the observed endothelial proliferation-to-differentiation switch in our study. bFGF is one of the proteins included in the antibody array kit used in this study, but its expression was not detectable in the secretome of CM of LS 174T cell lines. Therefore, it is likely that bFGF acts upstream instead of downstream of NO in the proliferation-to-differentiation switch in our study.

Since EA treated with LS 174T-PrP CM released the highest amount of NO, NO could elicit a cytostatic rather than cytotoxic effect (*Cartwright, Johnstone & Whitley, 2000*). This could explain the observed constant repression of proliferation rate of EA treated with LS 174T-PrP CM across 24, 48 and 72 h. Also, the role of NO in attenuating endothelial migration by reducing MMP-2 and MMP-9 secretions as seen in our study, is consistent with a previous study which showed that the transfection of ECs with endothelial NO synthase attenuated its migration (*Chen & Wang, 2004*). Besides, it was shown that that secretion of NO disrupted the adhesion of tumor cells to EC by downregulating the expression of cell adhesion molecules like integrins, cadherins, and secretins (*Lu et al., 2014*). These CAMs were not part of the antibody array and therefore we could not assess whether NO may affect the expression of CAMs in our study. Nevertheless, taken together, increased NO secretion might act as a crucial molecular signal in terminating EC proliferation and initiating the formation of vascular tubes.

Partial secretome profiling revealed distinctive angiogenesis-related protein secretion patterns from LS 174T-γ-Syn and LS 174T-PrP, as compared to LS 174T cells. Three pro-angiogenic factors, CXCL16, IGFBP-2 and amphiregulin, were downregulated

in the LS 174T-γ-Syn and LS 174T-PrP secretome. These factors have been shown to promote EC proliferation, migration and differentiation (*Zhuge et al., 2005*; *Bordoli et al., 2011*; *Das et al., 2013*). Apart from that, the secretion of VEGF and IL-8, widely recognized pro-angiogenic factors (*Waugh & Wilson, 2008*; *Moens et al., 2014*), were slightly and drastically elevated, respectively, in LS 174T-γ-Syn and LS 174T-PrP CM. However, EA treated with these CM showed reduced angiogenic responses. Similar trend has been observed previously for IL-8, suggesting that its effects could be endothelial cell-specific and increased secretion of IL-8 secretion might not guarantee an enhanced angiogenic response (*Wu, Lin & Chen, 2005*). Reduced secretion of DPPIV and TIMP-4 in LS 174T-γ-Syn and LS 174T-PrP CM, angiogenic factors that promote EC migration (*Ghersi et al., 2001*; *Fernández & Moses, 2006*), could also cause reduced EA migration in our study. Of note, secretion of uPA and angiogenin was shown to be significantly increased from LS 174T-PrP. The uPA/uPAR system is one of the systems that play an imperative role in angiogenesis, especially in the degradation of ECM (*Falchetti et al., 2008*; *Raghu et al., 2010*). The presence of uPA inhibitors—thrombospondin-1 and thrombospondin-2 (*Campbell et al., 2010*) secreted from LS 174T-PrP, could dampen the pro-migration effect of uPA and uPA/uPAR-mediated pro-angiogenic effects of angiogenin. Hence, this might explain why EA cells treated with LS 174T-PrP have reduced proliferation, invasion and migration rates.

Despite of the overall reduced angiogenic response from EA, the results showed successful EC tube formation. Most studies explained the pro- or anti-angiogenic effects of a certain factor in isolation, as ECs were only exposed to a certain factor alone in the experimental settings. As the secretomes of LS 174T-γ-Syn and LS 174T-PrP consist of multiple angiogenic factors, it is difficult to pinpoint in this study which of the secreted factor(s) could have been responsible for our observations. Also, the antibody array used in this study only represents a partial angiogenic secretome of the LS 174T cell lines. We speculate that there could be more signaling pathways being modulated following γ-Syn or PrP$^C$ overexpression in LS 174T. For example, soluble tissue factor, a factor not included in the antibody array, could exert a similar paradox as observed in this study, by promoting the migration and tubule formation of ECs, but not cell proliferation (*He et al., 2008*). Nevertheless, results of this array could provide valuable information that could lead to further investigation.

## CONCLUSION

Taken together, we demonstrated that two neuronal proteins, γ-Syn and PrP$^C$, which have been documented to be overexpressed in colorectal cancer cells, have the ability to activate EC angiogenic response. Specifically, these two proteins could be possibly involved in colorectal cancer-induced angiogenesis by inducing an endothelial proliferation–differentiation switch, which stops ECs from proliferating and foster ECs to differentiate and establish tube-like structures. NO could be the main factor in governing this transition, together with other angiogenesis-related proteins regulated under the influence of overexpressed γ-Syn or PrP$^C$.

### Funding

This work was supported by grants from the UTAR Research Fund—IPSR/RMC/
UTARRF/C210/S1 and IPSR/RMC/UTARRF/2012-C2/G05. The funders had no role in
study design, data collection and analysis, decision to publish, or preparation of the
manuscript.

### Grant Disclosures

The following grant information was disclosed by the authors:
UTAR Research Fund: IPSR/RMC/UTARRF/C210/S1 and IPSR/RMC/UTARRF/2012-C2/G05.

### Competing Interests

The authors declare that they have no competing interests.

### Author Contributions

- Sing-Hui Ong conceived and designed the experiments, performed the experiments,
  analyzed the data, contributed reagents/materials/analysis tools, prepared figures and/or
  tables, authored or reviewed drafts of the paper, approved the final draft.
- Kai-Wey Goh performed the experiments, contributed reagents/materials/analysis tools,
  authored or reviewed drafts of the paper, approved the final draft, creation of LS
  174T-gSyn cell line.
- Cornelius Kwang-Lee Chieng performed the experiments, contributed reagents/
  materials/analysis tools, authored or reviewed drafts of the paper, approved the final
  draft, creation of LS 174T-PrP cell line.
- Yee-How Say conceived and designed the experiments, analyzed the data, prepared
  figures and/or tables, authored or reviewed drafts of the paper, approved the final draft.

### Data Availability

  Ong, Sing-Hui; Goh, Kai-Wey; Chieng, Cornelius Kwang-Lee; Say, Yee-How (2018):
PrP GSyn Angiogenesis Paper YHSay Raw Data.xlsx. figshare. https://doi.org/10.6084/m9.
figshare.5853198.v1.

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
