# Peer review of "Cellular prion protein and γ-synuclein overexpression in LS 174T colorectal cancer cell drives endothelial proliferation-to-differentiation switch"

_PeerJ, doi:10.7717/peerj.4506_

## Round 0.1 · original submission · Minor Revisions

Please address all critical points raised by the reviewers. Note that Reviewer 2 has placed all their comments in the attached document

Reviewer 1 ·

Basic reporting

General comments

There are quite a few grammatical errors throughout the paper. These errors, especially those in the abstract make it difficult to follow the paper at times.



Minor errors

1. Line 291 typo: astrocyte
2. Line 234: Specify which software
3. The "internal control" on figure 1D needs to be clarified
4. Figure 2B is too wide
5. Figure 2D's low resolution makes it difficult to see the scratch assay
6. No scale bar for figure 2D
7. The Y axis for the figure 2D should probably be Time (hr)
8. Figure 2C has two graphs which share the same Y axis title "Conditioned media". It maybe better to title them " conditioned media from LS cells" or something like that to differentiate the two.
9. No Scale bar in Figure 3 images
10. Adjust the size of Figure 3 graphs
11. No Scale bar for Figure 4 images

Experimental design

The experimental design is appropriate.

Validity of the findings

Statistics mentioned are limited to identifying if the data have a statistical p value of less than 0.05. The P values should be mentioned and the data whose p value is below 0.01 or 0.001 can be highlighted

Comments regarding the discussion

While the cited Bussolino, Mantovani & Persico review article indicates that the angiogenesis involves changes in morphology, it does not mention EA cell line used in the manuscript. Is there any other reference that can show evidence that EA tube formation can be used as a model for the formation of new vasculature?

Is there any evidence that NO levels may affect the proteins detected in the secretome (Figure 5)? Or are they independent

---

## Round 0.2 · accepted · Accept

Thank you for addressing critical points raised by the reviewers.